# COMFYGEN: PROMPT-ADAPTIVE WORKFLOWS FOR TEXT-TO-IMAGE GENERATION

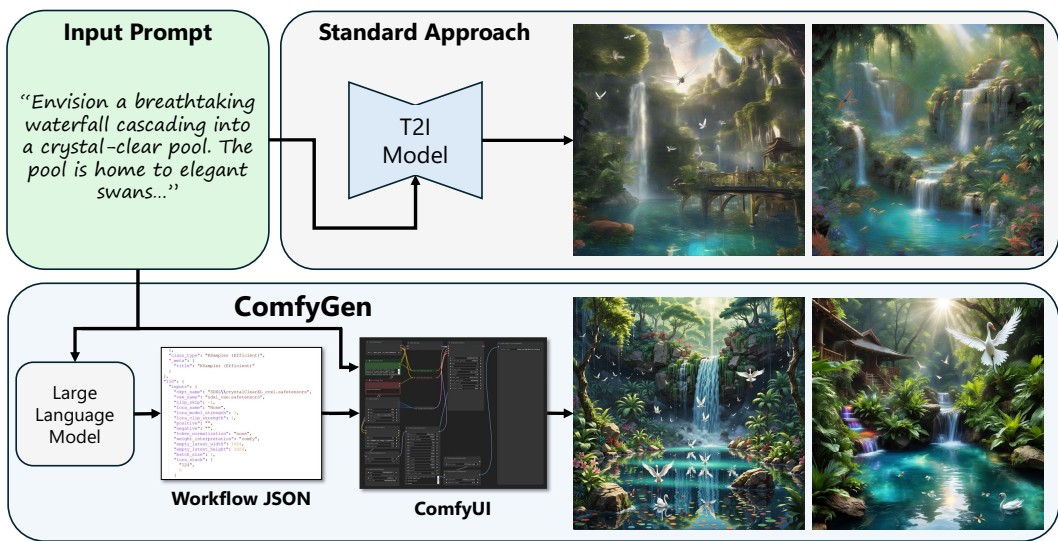

Figure 1: The standard text-to-image generation flow (top) employs a single monolithic model to transform a prompt into an image. However, the user community often relies on complex multi-model workflows, hand-crafted by expert users for different scenarios. We leverage an LLM to automatically synthesize such workflows, conditioned on the user's prompt (bottom). By choosing components that better match the prompt, the LLM improves the quality of the generated image.

## ABSTRACT

The practical use of text-to-image generation has evolved from simple, monolithic models to complex workflows that combine multiple specialized components. While workflow-based approaches can lead to improved image quality, crafting effective workflows requires significant expertise, owing to the large number of available components, their complex inter-dependence, and their dependence on the generation prompt. Here, we introduce the novel task of *prompt-adaptive workflow generation*, where the goal is to automatically tailor a workflow to each user prompt. We propose two LLM-based approaches to tackle this task: a tuning-based method that learns from user-preference data, and a training-free method that uses the LLM to select existing flows. Both approaches lead to improved image quality when compared to monolithic models or generic, prompt-independent workflows. Our work shows that prompt-dependent flow prediction offers a new pathway to improving text-to-image generation quality, complementing existing research directions in the field.

## 1 INTRODUCTION

As the field of text-to-image generation (Rombach et al., 2022; Ramesh et al., 2021) matures, researchers and practitioners shift from simple, monolithic workflows to more complex ones. Instead of relying on a single model to produce an image, those advanced workflows combine a variety of components, or blocks, designed to enhance the quality of the generated image (AUTOMATIC1111, 2022; Zhang, 2023; comfyanonymous, 2023). These components may include fine-tuned versions

of the generative model, large language models (LLMs) for refining the input prompt, LoRAs (Luo et al., 2023; Ryu, 2023) trained to correct poorly generated hands or to introduce specific artistic styles, improved latent decoders for creating finer details, super resolution blocks, and more.

Importantly, effective workflows are prompt-dependent. The choice of blocks often depending on the text prompt and the content of the image being created. For example, a workflow aimed at emulating nature photographs may elect to use a model fine-tuned for photorealism, while workflows focused on generating human images often contain the term "bad anatomy" as a negative prompt or leverage specific super-resolution models that also correct distorted facial features, such as the eyes. Due to the richness of available blocks and complexity of workflows, building a well-designed workflow often requires considerable expertise.

In this work, we propose to *learn* how to build text-to-image generation workflows, conditioned on a user prompt. Specifically, we propose to leverage an LLM to take as input a prompt describing an image, and output a workflow that is specifically tailored to that prompt. Below, we outline two approaches to achieving this goal. The prompt-specific workflow can then be used to synthesize images for that prompt, resulting in improved quality compared to using fixed base models or popular human-crafted workflows. Importantly, using an LLM enables the model to leverage its extensive prior knowledge to parse the prompt and match its content to the most appropriate blocks.

To represent flows, we build on ComfyUI (comfyanonymous, 2023), a widely used tool that stores workflows as JSON files, which can be easily parsed by recent LLMs. The popularity of ComfyUI also provides access to multiple human-created workflows, which we then augment to create a more diverse training set. To teach the LLM the link between flow components and image quality, we collect 500 diverse prompts from human users.[1] These prompts are used for generating images using each workflow in our training set, and the results are scored by an ensemble of aesthetic predictors and human preference estimators (Kirstain et al., 2023; Xu et al., 2024; Wu et al., 2023b). This process effectively creates a training set composed of triplets of (prompt, flow, score).

We then consider two approaches for matching flows to novel prompts. In the first, we leverage a closed-source LLM, and provide it with a table of flows and their scores across a closed-set of categories automatically derived from our training prompts. This table serves as a context for a followup request, where we ask the LLM to select the flow that is most suitable for a novel prompt. In the second approach, we fine-tune an open LLM (Dubey et al., 2024) so that, given a prompt and an ensemble score, it predicts the flow that achieved that score. During inference, we provide the LLM with an unseen prompt and a target score and ask it to provide us with an appropriate workflow. We name these approaches ComfyGen-IC and ComfyGen-FT respectively. The design choices behind each approach and their motivations are discussed below.

We compare our prompt-adaptive approach against several baselines, including: (1) single-model approaches (the baseline SDXL model (Podell et al., 2024), popular fine-tunes, and a DPO-optimized version (Rafailov et al., 2024; Wallace et al., 2024)), and (2) prompt-independent, popular workflows. ComfyGen outperforms all baselines on both human-preference and prompt-alignment benchmarks, highlighting the benefit of prompt-dependent flows.

Finally, we analyze the workflows selected by our method, demonstrate their relation to the domains represented in the input prompts, and investigate the scaling behaviors of our model.

## 2 RELATED WORK

**Improving Text-to-image generation quality.** With the growing popularity of text-to-image diffusion models (Rombach et al., 2022; Nichol et al., 2021; Ramesh et al., 2022), a range of works sought to improve the visual quality of their outputs, and their alignment to human preferences.

One approach is to fine-tune pretrained models using curated, high quality datasets or improved captioning techniques (Dai et al., 2023; Betker et al., 2023; Segalis et al., 2023). Instead of collecting data, a range of works use reward models (Kirstain et al., 2023; Wu et al., 2023b; Xu et al., 2024; Lee et al., 2023) to guide text-to-image generation. This can be done using reinforcement-learning (Black et al., 2024; Deng et al., 2024; Fan et al., 2024; Zhang et al., 2024). However, these methods can be computationally expensive and struggle to generalize effectively. As an alterna-

---

[1]Sampled from `https://civitai.com/` after filtering out NSFW content.

tive, the model can be fine-tuned using differentiable rewards (Clark et al., 2024; Prabhudesai et al., 2023; Wallace et al., 2024). Instead of tuning the model directly, one can also use reward models to explore the diffusion input-noise space (Eyring et al., 2024; Qi et al., 2024), finding seeds for which the output is of higher quality. Finally, some approaches leverage self-guidance (Hong et al., 2023) or frequency-based feature manipulations (Si et al., 2024; Luo et al., 2024) to drive the model towards more detailed and sharper outputs.

Our work proposes a new, orthogonal path to improving image quality. Instead of modifying the diffusion model or intervening in its sampling process, we use reward models to better match workflow components to a given prompt, aligning the entire pipeline towards human preferences.

**LLM-based tool selection and Agents** Recent advancements in large language models have demonstrated significant improvements in reasoning abilities and their capacity to adapt to novel content and tasks. This adaptability can be achieved through efficient fine-tuning methods, but more commonly simply through zero-shot prompting or in-context learning.

Building on these capabilities, a range of works proposed to leverage LLMs for tasks beyond text generation. A common line of work aims to equip the LLM with external tools (Schick et al., 2024), either through appropriate API tags within the generated text (Schick et al., 2024), by providing in-context API documentations (Wang et al., 2024; Surís et al., 2023), model descriptions (Shen et al., 2024) and code samples (Gupta & Kembhavi, 2023), or by retrieving models from a pre-defined collection. (Wu et al., 2023a). Such tools are often referred to as LLM agents, and their latest variants are often equipped with components such as memory mechanisms, retrieval modules or self-reflection and reasoning steps, all aimed at improving their overall performance.

Our work can similarly be viewed as an agent, as it employs an LLM to directly select and connect external tools. Here, we focus on the novel task of prompt-adaptive pipeline creation, and on tapping this under-explored path to improving the quality of downstream generations.

**Worfklow generation** An emerging trend in machine learning research is the use of compound systems, where multiple models are used in collaboration to achieve state-of-the-art results. These systems have been successfully used across various domains, ranging from coding competitions (AlphaCode Team, 2024) to olympiad-level problem solving (Trinh et al., 2024), medical reasoning (Nori et al., 2023) and video generation (Yuan et al., 2024). Crafting such compound systems can be a daunting task, as the components must be carefully selected and their parameters tuned to perform well in tandem, rather than optimized on each individual step of the task (). To address this, recent approaches have proposed optimization-based frameworks that tune pipeline parameters for improved end-to-end performance (Khattab et al., 2023), or even optimize the connections within a graph representing the components of a complex system (Zhuge et al., 2024).

Our work similarly tackles the task of pipeline generation. Here, we focus on text-to-image models, and demonstrate that their performance can be enhanced by designing compound pipelines that depend on the user's prompt.

## 3 METHOD

Given an input prompt describing an image, our goal is to match it with an appropriate text-to-image *workflow*, leading to improved visual quality and prompt alignment. We hypothesize that effective workflows will depend on the specific topics present in the prompt. Therefore, we propose to tackle this task by leveraging an LLM that can concurrently reason over the prompt and identify these topics, while also serving as a means to directly select or synthesize the new flow.

In the following section, we provide a detailed description of ComfyUI and our method, focusing on our training data, as well as our retrieval-based and score-based tuning approaches.

### 3.1 COMFYUI

Our work focuses on ComfyUI, an open-source software for designing and executing generative pipelines. In ComfyUI, users create pipelines by connecting a series of blocks that represent specific models or their parameter choices. In fig. 2a, we show a simple example ComfyUI pipeline. This pipeline includes blocks for loading a model, specifying prompts and latent dimensions, a sampler,

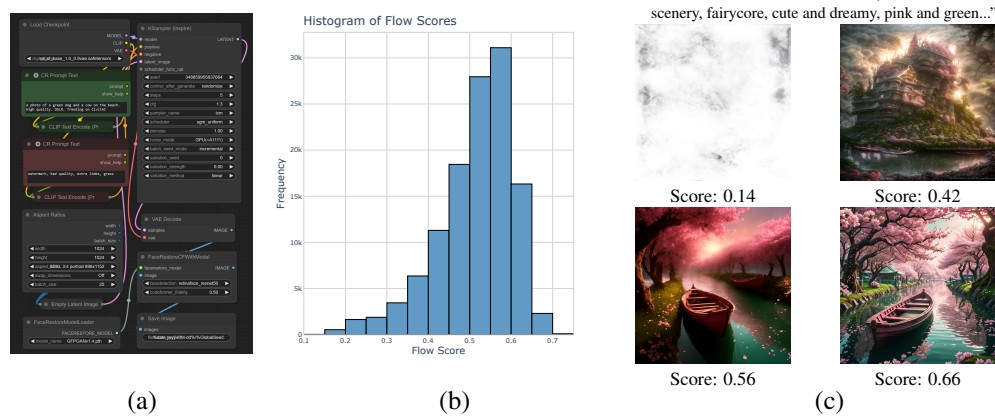

Figure 2: **(a)** A simple ComfyUI pipeline using a base model and a face restoration block, as well as both a positive and a negative prompt. **(b)** Distribution of scores for the prompt, flow pairs in our training set. **(c)** Example images produced for the same prompt by flows with different scores. A higher score typically correlates with more detailed and vibrant results, and fewer artifacts.

a VAE decoder, and a face restoration upscaling model. More complex pipelines may involve additional components like LoRAs (Ryu, 2023) or embeddings (Gal et al., 2022), ControlNets (Zhang et al., 2023), IP-Adapters (Ye et al., 2023), blocks that re-write and enhance the input prompt, and more. In many cases, complex blocks are introduced into ComfyUI through user-created extensions, which are then shared across the community.

Importantly, each ComfyUI pipeline can be exported to a JSON file which outlines both the graph nodes and their connectivity. ComfyUI's standard JSON format also contains UI information, such as the position and color of the blocks. We use the simpler API version which excludes this UI-specific information. Moreover, the API format of the flow can be used to trigger novel generations without using the UI, allowing us to automate much of our process.

We note that the concurrent work of Xue et al. (2024) also leverages ComfyUI pipelines. However, their work focuses on using ComfyUI as a test bed for exploring the stability of collaborative workflow generation approaches. Hence, their evaluation focuses on examining the rate at which ComfyUI-compliant workflows are created. In contrast, we focus on learning to tailor specific workflows to a user's prompt, with the aim of improving downstream generation quality.

## 3.2 TRAINING DATA

As a starting point, we collect a set of approximately 500 human-generated ComfyUI workflows from popular generative-resource-sharing websites such as Civitai.com. We limit ourselves to text-to-image workflows, flitering out video generation flows, and flows that take a control image as an input. We further discard highly complex flows, whose JSON representations often span tens of thousands of lines. Finally, we discard flows that use community-written blocks appearing in fewer than three flows. This leaves us with a small set of 33 flows, which we augment by randomly switching diffusion models (see supplementary for list), LoRAs and samplers, or changing parameters like the guidance scale and number of steps. In total, this process resulted in 310 distinct workflows.

Recall that our goal is to predict *effective* flows for a given prompt, which will enhance the quality of the generated output. To achieve this, we need a way to assess workflow performance. To do so, we begin by collecting a set of 500 popular prompts from Civitai.com and using them to synthesize images with each flow in our dataset. These images are then scored using an ensemble of quality prediction models (LAION Aesthetic Score (Schuhmann et al., 2022), ImageReward (Xu et al., 2024), HPS v2.1 (Wu et al., 2023b), and Pickscore (Kirstain et al., 2023)). We standardize the outputs of these models so that they are of approximately the same scale, and sum them up, assigning higher weights to models that better align with human preferences. This process yields a single scalar score for each prompt and flow pair, where higher scores typically correlate with better image quality. Figure 2b,c shows the distribution of scores in our data set, along with visual examples of images created with the same prompt, across a range of scores.

Our final dataset consists of triplets of prompt, workflow, and score. We use these to implement both the in-context and the fine-tuning based approaches detailed below.

### 3.3 COMFYGEN-IC

As a first approach to providing prompt-dependent flows, we look to in-context based solutions that leverage a powerful, closed-source LLM. To do so, we first need to provide the LLM with some knowledge about the quality of results produced by each flow. We thus start by asking the LLM to come up with a list of 20 labels which will best fit our 500 training prompts. These include object-categories ("People", "Wildlife"), scene categories ("Urban", "Nature") and styles ("Anime", "Photo-realistic"). A complete list of the labels is provided in the supplementary. With these labels in hand, we can now calculate the average quality score of images produced by each flow across all prompts belonging to a specific label. Repeating this for all flows and all labels gives us a table of flows and a measure of their performance across all 20 categories.

Ideally, we would have liked to provide the LLM with the full JSON description of the flows, allowing it to learn the relationships between flow components and downstream performance on specific categories. Unfortunately, the flows are too lengthy to fit more than a handful into the context window of most LLMs. Hence, our table contains only flow identities, and we simply ask the LLM to choose the flow that it believes will perform best on a given, unseen prompt.

All in all, this approach provides us with a classifier capable of parsing new prompts, breaking them down into relevant categories, and selecting the flow that best matches these categories. We name this variation ComfyGen-IC.

### 3.4 COMFYGEN-FT

As an alternative approach, we can fine-tune an LLM to predict high-quality workflows for given prompts. One way to approach this problem could be to simply fine-tune the LLM so that, given an input prompt provided in-context, it would need to predict the flow that achieved the highest score for that prompt. However, this approach has several drawbacks: it significantly reduces the number of training tokens, using only one flow per prompt instead of all 310; it's more sensitive to randomness in our data creation process, such as the seed chosen for each generated image; and it doesn't allow learning from negative examples, which could help the model identify ineffective flow components or combinations.

Instead, we propose an alternative fine-tuning formulation where the LLM takes as its context both the prompt and a score. The model is then tasked with predicting the specific flow that achieved the given score for the corresponding prompt. This approach tackles the drawbacks of the best-scoring-flow method. First, it greatly increases the number of tokens available for training by utilizing all available data points, not just the highest-scoring ones. Second, it reduces the impact of random fluctuations by considering a wider range of scores and their associated flows. Third, it allows the model to learn from negative examples, i.e., flows that achieved low scores for a given prompt. At inference time, we can simply provide the LLM with a prompt and a high score and have it predict an effective flow for our prompt. We name this variation ComfyGen-FT.

### 3.5 IMPLEMENTATION DETAILS

We implement ComfyGen-IC using Claude Sonnet 3.5, and ComfyGen-FT on top of pre-trained Meta Llama3.1 8B and 70B checkpoints (Dubey et al., 2024). Unless otherwise noted, all ComfyGen-FT results in the paper use the 70B model with a target score of $0.725$. In all cases, we finetune for a single epoch using a LoRA of rank 16 and a learning rate of $2e-4$. Additional details are in the supplementary.

## 4 EXPERIMENTS

We begin by showcasing images generated with our approach across a range of prompts, including subject-focused, photo-realistic imagery, as well as artistic or abstract creations. These are shown in fig. 3, with additional large-scale figures in the supplementary.

Next, we compare the images produced by our approach with those generated by a series of baselines. We consider two types of alternative approaches: (1) Fixed, monolithic models, where we

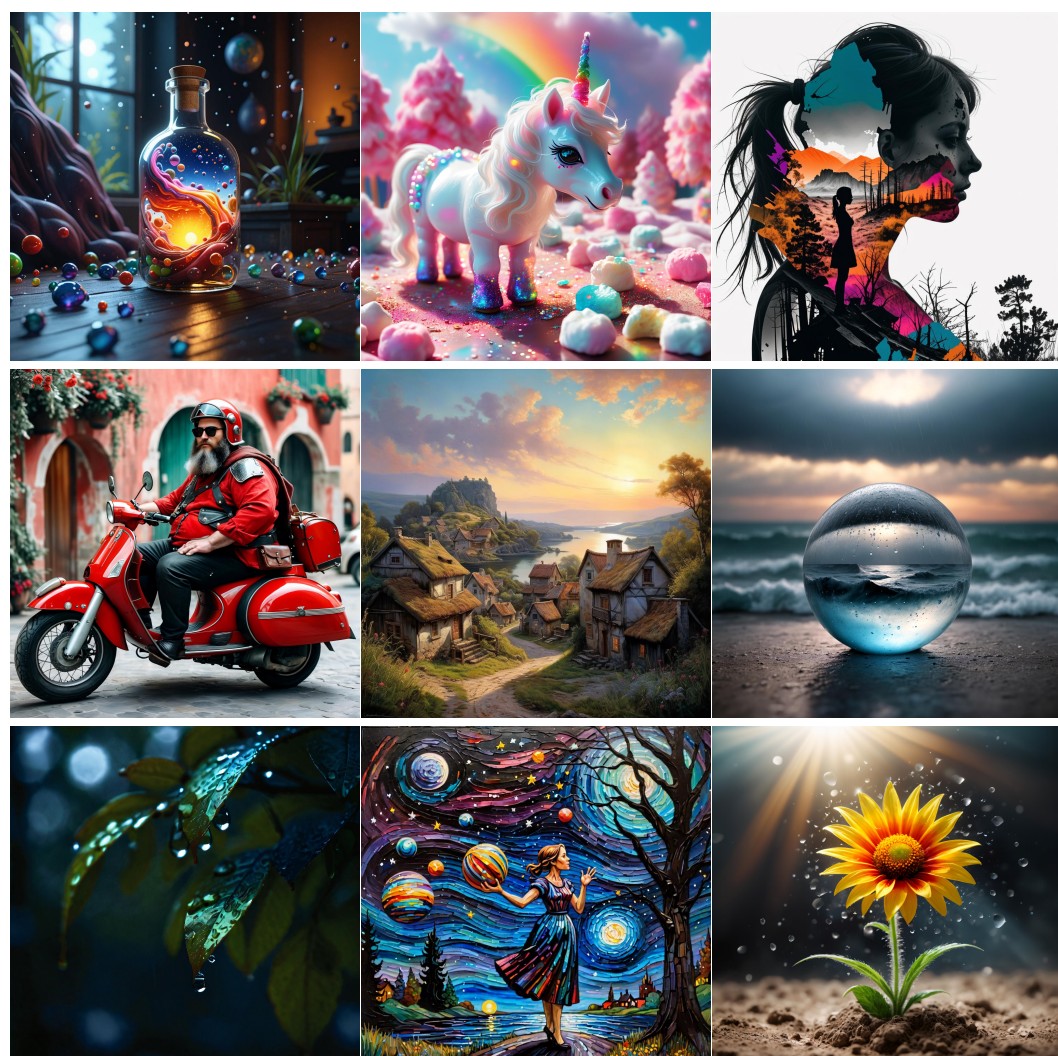

Figure 3: Our method can generate higher quality images across diverse domains and styles. Prompts are available in the supplementary.

simply use the prompts to directly condition a pre-trained diffusion model. (2) Generic workflows, where we use the same workflow to generate all images, regardless of the specific prompt.

For (1), we consider the original SDXL model (Podell et al., 2024) and its two most popular (most downloaded on CivitAI) fine-tuned variations: JuggernautXL and DreamshaperXL. We further consider a version of SDXL fine-tuned with DPO (DPO-SDXL, (Wallace et al., 2024)). For (2), we selected the two most popular flows (based on download counts) from our training corpus. These flows use SSD-1B (Gupta et al., 2024) and Pixart-$\Sigma$ (Chen et al., 2024) respectively.

We evaluate our models and the baselines on two fronts. First, we use the GenEval benchmark (Ghosh et al., 2024), which uses object detection to assess generative models across prompt-alignment tasks like single-object generation, counting, and attribute binding. Quantitative results are reported in table 1 with qualitative samples shown in fig. 4. Our tuning-based model outperforms all baselines, despite only using human preference scores during training. The in-context approach underperforms. We hypothesize that it suffers due to GenEval's short, simplistic prompts, which typically only describe a few objects in one or two words each. This challenges the LLM's ability to match the prompts with labels, and performance degrades.

To better evaluate the visual quality of our images, we turn back to CivitAI, and sample 500 prompts from the 10, 000 highest-ranked images on the website, after filtering out prompts used for training

| Model | Single object | Two object | Counting | Colors | Position | Attribute binding | **Overall** |
|---|---|---|---|---|---|---|---|
| Single Model - SDXL | 0.98 | 0.74 | 0.39 | 0.85 | 0.15 | 0.23 | 0.55 |
| Single Model - JuggernautXL | **1.00** | 0.73 | 0.48 | 0.89 | 0.11 | 0.19 | 0.57 |
| Single Model - DreamShaperXL | 0.99 | 0.78 | 0.45 | 0.81 | **0.17** | 0.24 | 0.57 |
| Single Model - DPO-SDXL | **1.00** | 0.81 | 0.44 | **0.90** | 0.15 | 0.23 | 0.59 |
| Fixed Flow - Most Popular | 0.95 | 0.38 | 0.26 | 0.77 | 0.06 | 0.12 | 0.42 |
| Fixed Flow - 2nd Most Popular | **1.00** | 0.65 | **0.56** | 0.86 | 0.13 | **0.34** | 0.59 |
| ComfyGen-IC (ours) | 0.99 | 0.78 | 0.38 | 0.84 | 0.13 | 0.25 | 0.56 |
| ComfyGen-FT (ours) | 0.99 | **0.82** | 0.50 | **0.90** | 0.13 | 0.29 | **0.61** |

Table 1: GenEval comparisons. ComfyGen-FT outperforms all baseline approaches, despite being tuned with human preference scores, and not strictly for prompt alignment.

SDXL   Juggernaut   DreamShaper   Flow 1   Flow 2   ComfyGen-IC   ComfyGen-FT

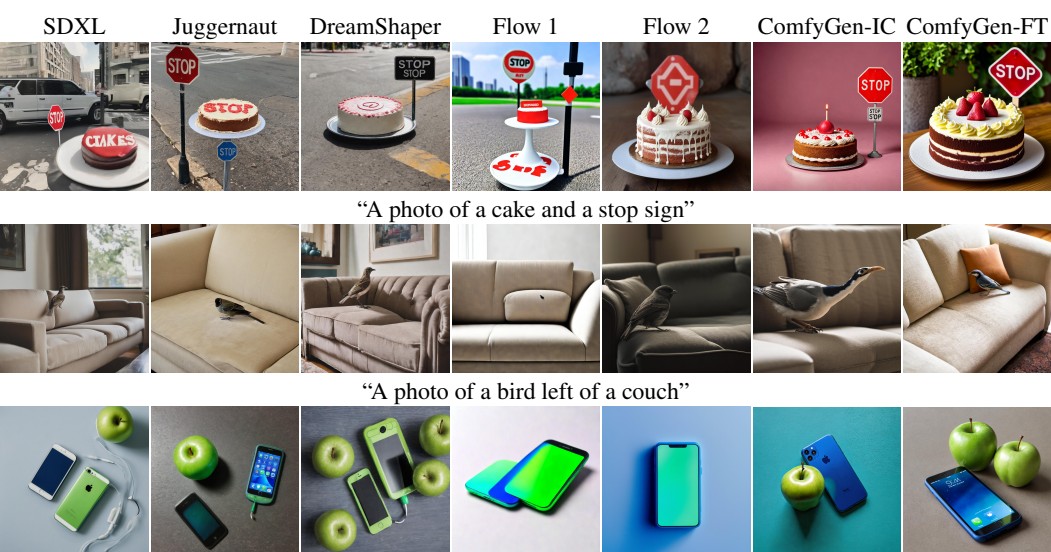

"A photo of a cake and a stop sign"

"A photo of a bird left of a couch"

"A photo of a blue cell phone and a green apple"

Figure 4: Qualitative results on GenEval prompts. ComfyGen shows better performance on multi-subject prompts, colorization and attribute binding, but may struggle with positioning.

our model and prompts that contain nudity or excessive violence. We assess the results both automatically — using HPS V2.0, a model not included in the weighted score used during training — and through a human preference study. For HPS, we follow Wallace et al. (2024); Qi et al. (2024) and perform a pair-wise comparison, each time pitting our method against a baseline using the same prompt. We report the fraction of prompts for which our approach received a higher score.

For the user study, we pit each version of our model against each baseline in a two-alternative forced-choice setup and report the fraction of times our model was preferred over each baseline. We sample roughly 20 prompts for each baseline and ComfyGen version pair, for a total of 231 questions. We collected a total of 682 responses from 35 users. More details are provided in the supplementary.

The results on the CivitAI prompts are shown in fig. 5, with visual samples for our approach and the 4 best baselines provided in fig. 6. Both of our approaches outperform all baselines, with notable improvement over simply using the baseline SDXL model. Curiously, we observe that some fine-tuned versions of SDXL are competitive with fixed flows, further emphasizing the importance of tailoring flows to specific use cases.

## 5 ANALYSIS

Having shown that our approach outperforms existing baselines, we next turn to analyzing its behavior. We examine three aspects of ComfyGen's performance: (1) the originality and diversity of the generated flows, (2) whether they show human-interpretable patterns, and (3) the effect of using the target score in the ComfyGen-FT prompts. The findings for these aspects are reported below.

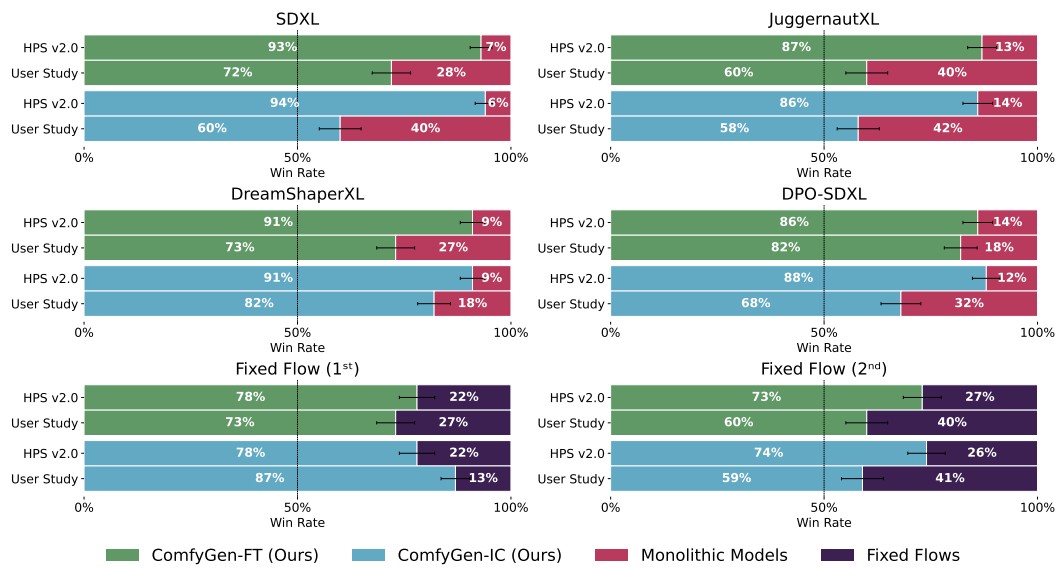

Figure 5: HPS V2.0 and User Study win rates. We compare each baseline against both ComfyGen-FT (green) and ComfyGen-IC (teal). ComfyGen variants are favored over all baselines.

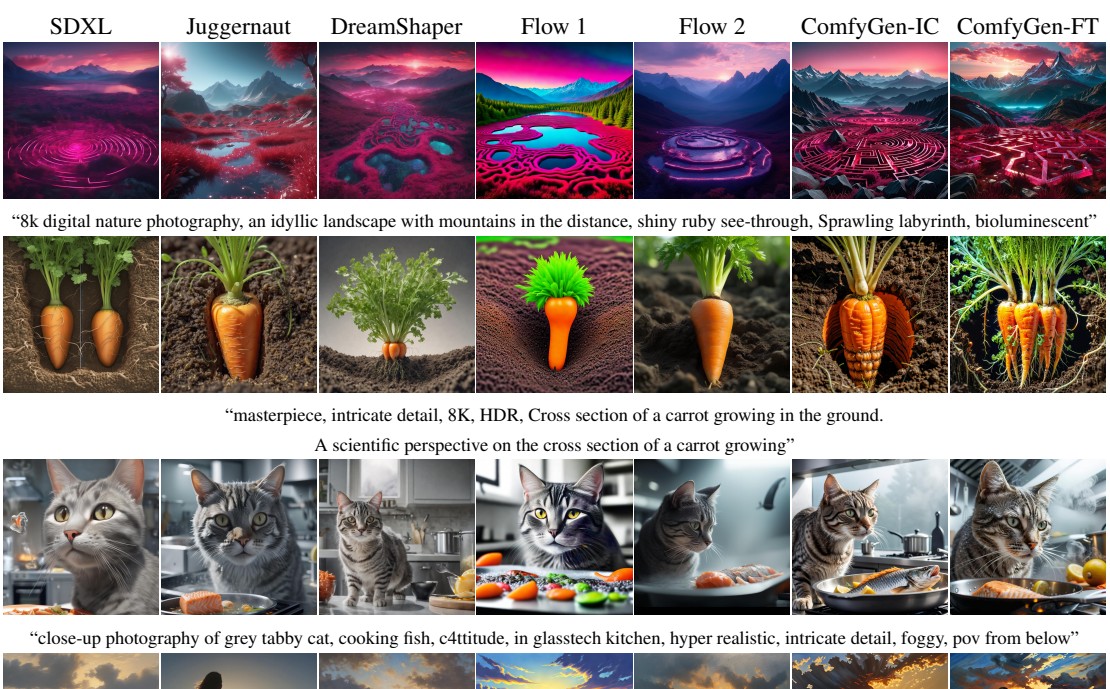

"8k digital nature photography, an idyllic landscape with mountains in the distance, shiny ruby see-through, Sprawling labyrinth, bioluminescent"

"masterpiece, intricate detail, 8K, HDR, Cross section of a carrot growing in the ground.
A scientific perspective on the cross section of a carrot growing"

"close-up photography of grey tabby cat, cooking fish, c4ttitude, in glasstech kitchen, hyper realistic, intricate detail, foggy, pov from below"

"oil painting, silhouette of a woman in the steppe wilderness, dramatic light, 34K uhd, masterpiece, high detail,
8k, intricate, detailed, high resolution, high res, high quality, highly detailed, Extremely high-resolution details, fine texture"

Figure 6: Qualitative results on CivitAI prompts. Please zoom in to better appreciate the differences.

## 5.1 ORIGINALITY AND DIVERSITY

We begin by assessing ComfyGen-FT's ability to generate novel flows by comparing its predictions for the 500 CivitAI prompts to the nearest neighbors in our training corpus. While ComfyGen-

IC is retrieval-based and has an expected similarity score of 1.0, ComfyGen-FT achieves 0.9995 similarity, indicating that at the scale of our model, there is little to no generation of unseen flows. This indicates that our fine-tuning approach has also learned to classify flows. However, we note that in contrast to ComfyGen-IC, it has learned so directly from the data with limited ad-hoc choices in the process, and indeed it outperforms ComfyGen-IC in most of our evaluations. Looking ahead, we hope that future methods will also be able to synthesize unseen flows with novel graph structures.

In terms of diversity, we observe that over 500 prompts, ComfyGen-IC makes use of 41 unique flows, while ComfyGen-FT uses 79, indicating a higher diversity. Recall also that our base set contained only 33 human created templates, which were then augmented through random parameter changes. Hence, both variations identified useful flows which differ from the initial human-created set. This suggests that more data or a more involved search over the input parameter space could yield more diverse outputs and possibly improved performance.

## 5.2 ANALYZING THE CHOSEN FLOWS

Next, we want to see whether we can identify any patterns in the chosen flows that would provide useful information about the strengths of existing models. Towards this goal, we want to see which models are popular across different categories, and which ones are especially prevalent for prompts within a particular category.

To identify models most strongly associated with specific labels, we parse all flows selected for our 500-prompt test set. We scan each flow for base models, LoRAs, and upscaling models, appending their names to a to a document corresponding to each label associated with the prompt that generated the flow. Then, we use TF-IDF (Sparck Jones, 1972) to rank the models across these label-documents. In table 2, we report the top-scoring model for each of four distinct labels, as well as the most common models across the entire flow set ("General").

We observe that, in many cases, the choices make intuitive sense. For example, the GFPGAN face restoration model is closely tied to the "People" category. Similarly, "Anime" prompts make more frequent use of models that better preserve human anatomy, or a LoRA tuned for an anime aesthetic. However, while such patterns exist in the data, the choices are not always intuitively clear. In the future, it may be beneficial to have the LLM explain the reasoning behind its component selections.

| Category | "People" | "Nature" | "Anime" | "Abstract" | General |
|---|---|---|---|---|---|
| Top Base Model | Proteus v3 | Stable Cascade | JibMixXL v9 "Better Bodies" | SDVN7-NijiStyleXL | crystalClearXL |
| Top LoRA | SDXL FaeTastic v24 | Add-Detail XL | AnimeTarot | LogoRedmond | MidJourney52 v1.2 |
| Top Upscaler | GFPGAN v1.4 | Real-ESRGAN | UltraSharp x4 | None | UltraSharp x4 |

Table 2: Top workflow components by TF-IDF scores for selected categories. In many cases, selections align with human intuition (*e.g.*, a face upscaling model is favored for the "People" category).

## 5.3 THE EFFECT OF TARGET SCORES

Recall that ComfyGen-FT was fine-tuned to predict a flow based on a given prompt and a target score. Here, we examine the performance of the model according to the target score provided at inference time. To do so, we repeat the CivitAI prompt experiments of section 4, while adjusting the target score used in our prompts. We evaluate both a model tuned from the Llama3.1 8B version and one from Llama3.1 70B. The quality of the generated images is assessed using HPS v2.0, and we report the average outcomes. The results are presented in fig. 7. For reference, we provide the scores of the baseline SDXL model, as well as ComfyGEN-IC. We additionally examine a scenario where instead of tuning the model to predict a flow given a prompt and a score, we simply tune it to predict the highest scoring flow ("Predict best").

We observe that the ComfyGen-FT model has indeed learned to associate the target scores with flows of varying quality. With an appropriate choice of score (near the top of the training score distribution), ComfyGen-FT achieves comparable performance to ComfyGen-IC. Notably, attempting to predict the best model instead of the score-based tuning leads to greatly diminished performance, highlighting the importance of our approach. We further note that both model sizes achieve comparable performance, hinting that we are far from saturating the capabilities of the models.

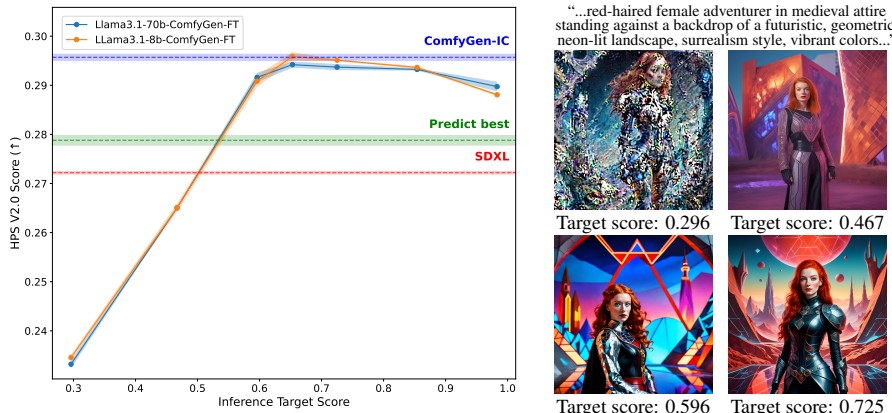

Figure 7: **(left)** Average HPS V2 score on CivitAI prompts as a function of the inference target score. The 8B and 70B variations of our model perform equally well, and significantly outperform the variation trained to predict the highest scoring flow (green). **(right)** Model outputs for the same prompt at different target scores.

Finally, it is important to note that our target scores were calculated using an ensemble of human-preference predictors, excluding HPS v2.0. Therefore, there should be no expectation for alignment between the values on the y-axis and the x-axis.

# 6    LIMITATIONS

While ComfyGen's prompt-dependent workflow approach demonstrates improvements over monolithic models and constant flows, it is not free of limitations. Our current model is limited to text-to-image workflows, and cannot address more complex editing or control-based tasks. However, this could potentially be resolved in the future through the use of vision-language models (VLMs).

Similarly, both of our approaches require us to generate images using a large number of flows. With typical generations taking an order of 15 seconds, even a modest set of 500 prompts and 300 flows requires a month of GPU time to create. Therefore, scaling up the approach would likely require significant computational resources or more efficient ways (e.g., Reinforcement Learning) to explore the flow parameter space.

Finally, each of our two methods has its own unique drawbacks. The fine-tuning approach cannot easily generalize to new blocks as they become available, requiring retraining with new flows that include these blocks. On the other hand, the in-context approach can be easily expanded by including the new flows in the score table provided to the LLM. However, this increases the number of input tokens used, making it more expensive to run and eventually saturating the maximum context length. We hope that these limitations can be addressed through more advanced retrieval-based approaches or through the use of collaborative agents.

# 7    CONCLUSIONS

We introduced the task of prompt-adaptive workflow generation, and presented ComfyGen - a set of two approaches that tackle this task. Our experiments demonstrate that such prompt-dependent flows can outperform monolithic models or fixed, user created flows, in a sense providing us with a new path to improving downstream image quality.

In the future, we hope to further explore prompt-dependent workflow creation methods, increasing their originality and expanding their scope to image-to-image or even video-related tasks. Perhaps in the future we could collaborate with the language model on the creation of such flows, providing it feedback through additional instructions or examples of outputs, thereby enabling non-expert users to further push the boundary of content creation.

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
