# OpenReview forum: "ComfyGen: Prompt-Adaptive Workflows for Text-to-Image Generation"
_ICLR.cc/2025/Conference — ICLR 2025 Conference Withdrawn Submission_

### Official Review · Reviewer_bBXt · 2024-10-23

**Soundness:** 3
**Presentation:** 3
**Contribution:** 3
**Rating:** 3
**Confidence:** 5

**Summary:**

This paper introduces the task of **prompt-adaptive workflow generation** for text-to-image models, addressing the complexity of creating effective workflows that combine multiple specialized components. The authors propose two LLM-based approaches: a tuning-based method that learns from user preferences, and a training-free method that selects existing workflows based on the prompt. Both methods improve image quality compared to monolithic models or prompt-independent workflows, highlighting the potential of prompt-dependent flow prediction as a new pathway for enhancing text-to-image generation.

**Strengths:**

1. This paper addresses an unexplored and intriguing problem: the automatic generation of ComfyUI workflows based on user-provided prompts, which holds significant potential for creative industries.

2. The prompt-adaptive workflow generation approach is novel, providing flexible solutions through both tuning-based and training-free methods.

3. The paper is clearly written and easy to follow, with a well-structured explanation of the problem and proposed solutions.

**Weaknesses:**

**1. Practicality Concerns**

The authors only trained the proposed model on 310 different text-to-image workflows and excluded workflows with uncommon blocks and complex workflows. This limited training data seems overly simplistic. The model may struggle to learn how to create complex workflows that incorporate a variety of custom blocks. As a result, the model might merely combine simple existing blocks and adjust parameters accordingly, which could significantly limit its practical effectiveness.

**2. Insufficient Validation**

While generating workflows from prompts is an interesting idea, it may not be the most optimal approach for improving text-to-image generation quality. Since this method indirectly improves image quality by refining the workflow rather than addressing the base model's inherent issues (e.g., attribute misbinding, counting errors), it may have limited effectiveness.
For instance, in Fig. 4 (third row), the proposed ComfyGen-FT fails to interpret the prompt "a green apple" and instead generates two apples. When compared to other methods that directly leverage LLMs to improve text-to-image generation quality, such as Lavi-Bridge [1], ELLA [2], and PixArt-Alpha [3], the proposed approach does not demonstrate a clear advantage. A more thorough analysis and comparison with these methods are recommended.

**3. Generalization Issue**

The proposed method struggles to generalize to new community blocks. Incorporation of these new blocks requires retraining the entire model, which poses limitations on its adaptability.

*_[1] Bridging Different Language Models and Generative Vision Models for Text-to-Image Generation. ECCV 2024._*

*_[2] ELLA: Equip Diffusion Models with LLM for Enhanced Semantic Alignment. Arxiv 2024._*

*_[3] PixArt-α: Fast Training of Diffusion Transformer for Photorealistic Text-to-Image Synthesis. ICLR 2024._*

**Questions:**

1. Does the proposed method have the capability to design new, previously unseen blocks based on the prompt, or is it limited to the blocks present in the training data?

2. Is there a mechanism in place to ensure that the generated workflows are correct and functional? Given the complexity of the connections between workflow blocks, any logical errors in the generated workflows could significantly impact the user experience.

3. Can the method integrate real-time feedback during workflow generation? For example, if the user wants to modify certain aspects of the generated workflow on the fly, is there a mechanism to incorporate this input dynamically?

4. How does the method handle complex workflows that involve many interdependent blocks? Given that the training data excluded complex workflows, how well does the model perform when tasked with generating workflows that require multiple custom or community blocks with intricate interdependencies?

---

### Official Review · Reviewer_foC6 · 2024-11-01

**Soundness:** 3
**Presentation:** 4
**Contribution:** 3
**Rating:** 5
**Confidence:** 4

**Summary:**

This paper introduces a new approach to text-to-image generation by dynamically adapting workflows to input prompts, rather than using monolithic models. The authors present two methods for workflow generation: ComfyGen-IC (an in-context prompt classifier) and ComfyGen-FT (a fine-tuned model for predicting workflows based on prompt and desired image quality score). By leveraging ComfyUI, which stores workflows as JSON files, the model maps prompts to workflows. The evaluations based on preference estimation models and user study are performed to validate the superiority of the proposed pipeline over fixed workflows and single-model baselines.

**Strengths:**

1.	Originality: The approach is novel in treating the text-to-image generation as a prompt-specific workflow generation task rather than using single, static models or workflows. By utilizing LLMs to dynamically create workflows based on prompts, this research introduces a new way to improve image quality.
2.	Quality: The quality of the work is strong, with well-designed experiments, clear distinctions between the ComfyGen-IC and ComfyGen-FT methods, and comparative baselines. Evaluation on both quantitative and qualitative benchmarks strengthens the work’s validity.
3.	Clarity: Overall, the paper is well-written, explaining both the advantages and challenges of prompt-adaptive workflows. Diagrams and tables are well-used to illustrate performance differences, and the methodology is well-detailed.
4.	Significance: The significance lies in expanding the text-to-image generation field by proposing a workflow-based approach that could potentially improve downstream applications. The work suggests a pathway for adaptive pipeline design in generative AI, a promising direction for improved customization in user-generated content.

**Weaknesses:**

1.	High Resource Requirements: ComfyGen-FT, in particular, requires extensive resources due to the need for numerous training data generated with different prompts and flows, with each generation taking around 15 seconds. This high computational overhead could be a limitation in scaling and adoption, given the user prompts and the needs are highly variant in real applications. In addition, during inference, the LLM part may take even longer to generate a JSON workflow (as discussed in the paper, there may be thousands of lines) than generating the images with diffusion models. The memory and computation overhead may make this pipeline not as applicable as expected.
2.	Workflow Validity and Robustness: ComfyGen-FT lacks explicit guarantees of workflow validity, meaning that some generated workflows may fail when applied. This issue could affect robustness and reliability, particularly when new workflows or blocks are introduced that require retraining or revalidation. Including a validity check could address this.
3.	Marginal Improvements: The performance improvement over some baselines (such as the 0.02 increase from 0.59 to 0.61 in GenEval scores) is modest. Considering the computational cost, it may not be sufficient to justify this approach compared to simpler models or retrieval-based systems without significant gains in other metrics like diversity or robustness.
4.	Diversity of Generated Flows: The diversity of generated workflows is limited, as noted by the authors themselves, with ComfyGen-IC appearing more like a retrieval-based method than a generation method. This limits the model's ability to create genuinely novel workflows beyond existing templates.

**Questions:**

1.	Could the authors clarify if any workflow validity checks are in place to ensure that generated flows will execute correctly in the ComfyUI environment? This would improve the method’s reliability.
2.	Given the high computational demands of ComfyGen-FT, would it be feasible to use a simpler classification model (8B model seems to be on par with 70B version) for selecting workflows? Since ComfyGen-IC functions similarly to retrieval, could lightweight models be considered?
3.	Does the limited diversity of workflows impact the model's generalizability for different prompt domains? It would be helpful to consider additional ways to expand flow diversity beyond parameter tuning, possibly by creating new flow templates or expanding the set of base workflows.
4.	The model’s component choices are not always intuitive, especially when associating specific LoRAs and upscalers with certain prompt types. Explainability mechanisms, such as scoring explanations, could help users understand why specific components are chosen for particular prompts, potentially increasing trust in the model's outputs. The authors can consider using more illustrations to show which model/LoRA/hyperparameters works better in which scenarios.

---

### Official Review · Reviewer_Gmfa · 2024-11-03

**Soundness:** 3
**Presentation:** 3
**Contribution:** 3
**Rating:** 6
**Confidence:** 3

**Summary:**

This paper proposes a prompt-adaptive workflow generation approach that utilizes LLMs to automate the workflows, thereby assisting users in their creative processes. The authors implement both retrieval-based and fine-tuning methods, building a comprehensive dataset consisting of <prompt, workflows, score>. The experiments demonstrate that the proposed method outperforms existing approaches. Additionally, a detailed user study is conducted, and the authors analyze the originality, diversity, and human-interpretable patterns of the method.

**Strengths:**

1. This paper is the first to identify the challenges associated with using workflows for creative tasks, addressing a valuable issue. And it  leverage LLMs to automate this process, which will significantly assist the community, particularly beginners, in enhancing their creative output.
2. The descriptions of ComfyGen-IC, ComfyGen-FT, and the dataset setup are clear and well-designed.
3. ComfyGen-FT demonstrates superior performance compared to other methods, including fixed models and commonly used workflows. Furthermore, ComfyGen-FT is trained on diverse scores while simultaneously learning from both positive and negative examples, which contributes positively to the model training process.
4. The results of the user study underscore the effectiveness of the proposed method, and the analysis section provides meaningful insights that enhance the understanding of this approach.

**Weaknesses:**

1. ComfyGen-IC is merely a retrieval-based method utilizing LLMs, and the diversity of workflows generated by both methods is somewhat limited.
2. The user study involved only 35 participants, which may limit the scalability of the findings.
3. There is a lack of discussion regarding the changes and corresponding explanations that would arise from using a higher score (rather than 0.725) in ComfyGen-FT during inference, as a higher score should correlate with better quality outputs.
4. In the qualitative results for ComfyGen-FT, the number of generated objects often appears incorrect, and there is insufficient explanation regarding the noted issues with "positioning" mentioned in the text.

**Questions:**

1. It would be valuable to explore how LLMs can be leveraged to generate a greater variety of new workflows, which would enhance the significance of this paper's contributions. This is my primary concern.
2. What are the results if ComfyGen-FT utilized higher scores during inference?
3. Why do the qualitative results for ComfyGen-FT show discrepancies in the number of generated objects, and what explanations can be provided for the identified issues related to "positioning"?

---

### Official Review · Reviewer_FdUY · 2024-11-04

**Soundness:** 2
**Presentation:** 3
**Contribution:** 2
**Rating:** 5
**Confidence:** 3

**Summary:**

The paper considers image generation with workflow systems such as ComfyUI. The main issue when using the workflow systems is the challenge of developing a valid flow that would lead to high quality image generation. The suggested approach leverages a LLM to generate a prompt conditioned flow. In particular, the paper is exploring two approaches: finetuning and in-context learning. Automatic and human evals show that the proposed approaches are outperforming fixed flows as well as off-the-shelf models.

**Strengths:**

The paper is well written and easy to follow.

The idea of automatizing flows with an LLM is interesting.

Qualitative results look nice.

**Weaknesses:**

The way the paper is presented it assumes that the reader is familiar with ComfyUI framework. However, the paper would be more complete when presenting more details about this framework, e.g., hos difficult it is to design a new flow in ComfyUI? As the reviewer never sued this environment, it is not easy to assess fully the value and complexity of the contribution.

The validation of the proposed method only includes a handful of models. Would it be possible to include more recent models such as FLUX and SD3? How strong the fixed flow baselines are? Overall, it is not easy to assess the significance of the reported results. It would also be nice to see more details about the validation – see some of the questions below.

**Questions:**

Line 140 has a missing citation.

Footnote #1 – please define NSFW

Would it be possible to see user study comparing ComfyGen-FT to ComfyGen-IC?

What are the metrics reported in Table 1? Could the authors clarify?

Adding more metrics would strengthen the validation too. How is the introduced method performing in terms of MSCOCO validation set FID? How about CLIP score?

---

### Official Review · Reviewer_U9YV · 2024-11-04

**Soundness:** 2
**Presentation:** 2
**Contribution:** 3
**Rating:** 5
**Confidence:** 4

**Summary:**

This paper introduces a novel approach to text-to-image generation by developing adaptive workflows that cater to specific user prompts, enhancing the quality of generated images. The authors propose two Large Language Model (LLM)-based methods: ComfyGen-IC, which uses an in-context learning approach, and ComfyGen-FT, which involves fine-tuning an LLM. These methods are designed to automatically synthesize workflows that better match user prompts, improving the standard approach of using a single, monolithic model.

**Strengths:**

The article defines an innovative text2image workflow that approaches text-to-image generation from a new perspective. Automated generation of JSON files utilizing the powerful structured data understanding capabilities of large language models. The results of comparative experiments and user testing show that the method proposed in this article has achieved better performance. From the current scale of data collection, this method will have a certain potential when more data pairs are screened out in subsequent work. The writing logic of the article is relatively clear.

**Weaknesses:**

1. The article as a whole tends to emphasize engineering design. If more theoretical analysis and qualitative discussions can be introduced (such as analyzing different data selection methods), it will enhance the value of the article.
2. The visualization of images in the article is insufficient. Providing more results for text2 images and the workflow proposed in the article can alleviate readers' concerns about the diversity and overall quality of the workflow, although the author has already discussed the issue of diversity.
3. The article contains a large amount of data filtering, and it is recommended to provide charts to more clearly describe the filtering rules and process.
4. The lack of a detailed description of evaluation indicators in the main text affects the understanding of experimental results.

**Questions:**

1.	It is not entirely clear how the author selected 500 workflows and what rules they followed.
2.	Does extending only 33 streams to 310 streams using a data augmentation approach reduce diversity?
3.	What is the definition of generation quality in the article? Should the definition be clearly described, and is there a comprehensive consideration of image quality rating and aesthetic rating?
4.	It is still uncertain which of the two ship images in Figure 2c has better quality.
5.	What basis do the selected 20 tag categories depend on?
6.	COMFYGEN-IC seems to allow LLM to directly infer statistical distributions, so would the lack of input on the specific content of JSON harm LLM's understanding?
7.	Can you develop a new structured representation that falls between text prompts and workflow JSON descriptions? This would pique my interest and potentially solve the issue of excessively long tokens. Of course, COMFYGEN-FT also solves the token issue.

---

### Author Response · Authors · 2024-11-14
**General response to reviewers**

We would like to thank the reviewers for providing feedback on our manuscript.

We were glad to see agreement that the task and approach presented in the paper are **innovative** or **valuable**, with reviewers highlighting both the **quality** of the results and the **potential impact** on the creative industries.

While we are withdrawing our work, we will incorporate some of your suggestions into the next version. Here, we want to offer a general response to some of the concerns raised by the reviewers.

---
---

One key point we wish to highlight is that research is an ongoing effort. Indeed, multiple reviewers raised ideas through which the method could be improved, but we do not think this is a flaw. Our paper introduces **a new task**, and we demonstrate that there is value in pursuing this task by showing a solution that **already improves over existing, relevant baselines.**

The fact that there are multiple clear paths for future improvement simply means that there is a lot of potential in this new field, and that it is likely to spur future research. Both of these are traits which we argue should be a reason for acceptance to a scientific conference, rather than for rejection.

Below, we address a few specific points raised by the reviewers.


**Limited output flow diversity**

Our paper introduces a new task, and we present two methods that show the concrete benefits of tackling this task. The limited diversity is an artifact of our current solution, but we do not believe this is a fundamental limit to future solutions for the same task. Indeed, our work and some of the reviewers already suggest ways in which this could be overcome. Importantly, **even at the current levels of diversity, our solutions meaningfully outperform all relevant baselines.**

**Inclusion of stronger models like SD3 or Flux.**

At the time of our submission, both models were fairly recently released and did not yet have an extensive ecosystem of specialized variants such as fine-tuned versions, improved VAEs or LoRAs. This limits the flows involving them to very few changes in components, which we believe would make for a less interesting scenario to investigate. For SDXL, this rich ecosystem already existed, allowing us to conduct more interesting experiments and compare against a wider range of existing scenarios (e.g., the SDXL-DPO model).

As more specialized Flux variants become available, so too will the benefit of extending our approach to Flux-based flows. Until then, we suspect that the inclusion of such blocks will mostly serve to show that Flux is better than SDXL.

**Developing a structured, intermediate representation**

This is a great direction to investigate, but we believe there is enough to explore in this direction that it would warrant its own paper.

**Marginal improvement over baselines (only 0.02 in GenEval)**

A score difference of 0.02 on GenEval is not marginal. As can be seen from the results, DPO-tuned models outperform the baseline SDXL by 0.04 points. We improve over this by another 50%. Another point of reference: The difference between SD2.1 and SDXL, as reported in the GenEval paper, is 0.05 points, smaller than the gap between our approach and SDXL. We will provide this reference point in the text of our next submission.

Note that we also outperform all baselines on human preference metrics.

**Concerns about incorrect number of objects in some generated images**

We are confused by these remarks. Indeed, as with all methods, our approach sometimes generates an incorrect number of objects. The quantitative scores on GenEval, an established benchmark with hundreds of prompts, show that it happens **less often** for our method compared to the majority of baselines, and that overall, our approach outperforms all other baselines in prompt alignment.

**Insufficient images in the paper**

Our paper and supplementary include images generated by our method for 55 different prompts, in line with the standard in the field (e.g., SD3 has ~65, SDXL-DPO has ~45, SDXL has ~40, Pixart-Sigma has ~35, DALL-E3 has fewer than 20).

**Comparisons to methods that replace the text-encoder with an LLM (Pixart-alpha, Lavi-Bridge, ELLA)**

Flow #2 uses Pixart-Sigma (see line 313), which is a more recent, better version of the Pixart series. Our approach outperforms it even in a flow based setup.

Pixart-Alpha has been evaluated on GenEval in the SD3 paper, and has an overall score of 0.48. This is below even the baseline SDXL.
Similarly, Lavi-Bridge and ELLA only have SD versions available, and we do not believe there is much insight to gain from adding comparisons against weaker, smaller-scale models.

Note that they also generally use more resources than our approach. ELLA report 14 days of 8*A100 GPUs, roughly double the compute budget of our data generation time + LLM tuning, after accounting for the fact we use an H100. Pixart-alpha use 753 A100 days.

---

### Note · Authors · 2024-11-14

I have read and agree with the venue's withdrawal policy on behalf of myself and my co-authors.